# Unraveling Lifelong Brain Morphometric Dynamics: A Protocol for Systematic Review and Meta-Analysis in Healthy Neurodevelopment and Ageing

**DOI:** 10.3390/biomedicines11071999

**Published:** 2023-07-14

**Authors:** Yauhen Statsenko, Tetiana Habuza, Darya Smetanina, Gillian Lylian Simiyu, Sarah Meribout, Fransina Christina King, Juri G. Gelovani, Karuna M. Das, Klaus N.-V. Gorkom, Kornelia Zaręba, Taleb M. Almansoori, Miklós Szólics, Fatima Ismail, Milos Ljubisavljevic

**Affiliations:** 1Radiology Department, College of Medicine and Health Sciences, United Arab Emirates University, Al Ain P.O. Box 15551, United Arab Emirates; 2Medical Imaging Platform, ASPIRE Precision Medicine Research Institute Abu Dhabi, Al Ain P.O. Box 15551, United Arab Emirates; 3Big Data Analytics Center, United Arab Emirates University, Al Ain P.O. Box 15551, United Arab Emirates; 4Department of Computer Science and Software Engineering, College of Information Technology, United Arab Emirates University, Al Ain P.O. Box 15551, United Arab Emirates; 5Internal Medicine Department, Maimonides Medical Center, New York, NY 11219, USA; 6Physiology Department, College of Medicine and Health Sciences, United Arab Emirates University, Al Ain P.O. Box 15551, United Arab Emirates; 7Neuroscience Platform, ASPIRE Precision Medicine Research Institute Abu Dhabi, Al Ain P.O. Box 15551, United Arab Emirates; 8Biomedical Engineering Department, College of Engineering, Wayne State University, Detroit, MI 48202, USA; 9Siriraj Hospital, Mahidol University, Nakhon Pathom 73170, Thailand; 10Provost Office, United Arab Emirates University, Al Ain P.O. Box 15551, United Arab Emirates; 11Obstetrics & Gynecology Department, College of Medicine and Health Sciences, United Arab Emirates University, Al Ain P.O. Box 15551, United Arab Emirates; 12Neurology Division, Medicine Department, Tawam Hospital, Al Ain, P.O. Box 15258, United Arab Emirates; 13Internal Medicine Department, College of Medicine and Health Sciences, United Arab Emirates University, Al Ain P.O. Box 15551, United Arab Emirates; 14Pediatric Department, College of Medicine and Health Sciences, United Arab Emirates University, Al Ain P.O. Box 15551, United Arab Emirates

**Keywords:** brain, nerve degeneration, brain diseases, neurodevelopmental disorders, ageing, regression analysis, atrophy, neuroimaging, meta-analysis, cognition

## Abstract

A high incidence and prevalence of neurodegenerative diseases and neurodevelopmental disorders justify the necessity of well-defined criteria for diagnosing these pathologies from brain imaging findings. No easy-to-apply quantitative markers of abnormal brain development and ageing are available. We aim to find the characteristic features of non-pathological development and degeneration in distinct brain structures and to work out a precise descriptive model of brain morphometry in age groups. We will use four biomedical databases to acquire original peer-reviewed publications on brain structural changes occurring throughout the human life-span. Selected publications will be uploaded to Covidence systematic review software for automatic deduplication and blinded screening. Afterwards, we will manually review the titles, abstracts, and full texts to identify the papers matching eligibility criteria. The relevant data will be extracted to a ‘Summary of findings’ table. This will allow us to calculate the annual rate of change in the volume or thickness of brain structures and to model the lifelong dynamics in the morphometry data. Finally, we will adjust the loss of weight/thickness in specific brain areas to the total intracranial volume. The systematic review will synthesise knowledge on structural brain change across the life-span.

## 1. Introduction

Brain structure continuously changes throughout life. In healthy individuals, age-related brain atrophy and neurodevelopment account for these changes. In patients with mental and psychological disorders, disease-related brain atrophy takes place. The atrophy can start in late adolescence and young adulthood and lead to abnormal brain development [1]. Differentiation between normal and abnormal structural changes remains a challenge [2,3,4]. The current study focuses on the age-specific anatomy of the brain and describes the structural evolution of the brain across the life-span. A descriptive model that recapitulates key features of brain development and ageing is a powerful medium for obtaining comprehensive knowledge on the abnormalities signalling neurodegeneration. The model is a potential tool assisting clinicians in the early diagnostics of dementia. Studies on the structural signs of both normal and abnormal brain development and ageing remain relevant today.

The necessity of continuous research on the aforementioned issue is paramount. Autism spectrum disorder puts a substantial socio-economic burden, and the average diagnostic delay after initial concerns does not differ among high-, medium- and low-income countries [5]. On average, autism incidence is around 80 cases per 100,000 children in developed economies. The UAE is in the top-10 list of countries with the highest autism rates, which reached 112.40 in 2021. The statistics on middle- and low-income countries is incomplete because of misreported cases [6].

Neurocognitive slowing is a typical functional outcome of normal brain ageing, while cognitive decline is a sign of cognitive deterioration. In high-income countries, the numbers of people with cognitive decline has been rising due to population ageing. The latest favourable trends in dementia incidence are typical for the Western countries. Meanwhile, wealthy countries of other regions show the opposite tendency [7]. It is expected that around 35.25 million people will be diagnosed with dementia in Asia by 2025, while 13.97 million people will be diagnosed in European countries [8]. Neurodegenerative diseases (ND) are among leading causes of life loss and disability among the elderly, and the number of deaths due to Alzheimer’s disease has risen disproportionally in comparison to the top attributed cases of mortality (e.g., heart disease, cancer, and strokes) [9]. Individual causes of dementia are hard to detect and predict. Therefore, the disease is commonly diagnosed at late stages after the prominent manifestation of intellectual decline.

The identification of infants and elderly at risk of neurological disorders is important for optimal disease management [10]. For this reason, scientists look for highly sensitive screening and diagnosing tools, which enable early therapeutic strategies and targeted interventions. According to the recent updates from molecular biology studies, genetic tests can detect over 500 neurodevelopmental diseases [11]. The test specificity is sufficient for distinguishing dementia variants [12,13]. Still, the available diagnostic methods possess the following limitations. First, genetic tests are invasive, labour intensive, and time consuming. In addition, neurologists face difficulties in choosing an appropriate genetic test and interpreting the laboratory results [14]. Second, neurobehavioral and cognitive questionnaires are relatively short. Usually, they are well-tolerated by examinees, but the sensitivity and specificity are low in the paediatric population and in adults. For example, the Hammersmith Infant Neurological Examination predicts adverse neurophysiological outcomes at 1 year with a sensitivity of 50–64% and specificity of 73–77% [15]. The sensitivity and specificity of differentiation among dementia phenotypes with cognitive tests is around 71.92% and 70.06%, correspondently [16,17]. Third, molecular imaging with positron emission tomography can reveal early metabolic changes that are not visible on CT and MRI scans. The latter mainly document the brain macrostructure [18]. Although nuclear imaging is highly sensitive, the examination is invasive. The exposure to radiation and possible allergic reactions are the drawbacks of radiotracer injection. Additionally, the number of PET scanners is insufficient to arrange the routine screening of patients at risk of dementia [19,20]. MRI is the method of choice for detecting structural abnormalities in the brain, identifying disease-specific diagnostic signs at late stages of neurodegeneration, and reflecting their functional outcomes [4,21,22]. The early diagnostics of dementia necessitate the quantitative analysis of MRI findings along with bioengenering technologies. The creation of population norms will advance these technologies, thus meeting the demands of neuroscience for early and reliable diagnostics.

Scientists fail to describe the exact pathophysiological mechanisms in which age-related brain atrophy contributes to malfunctioning of the nervous system. Many publications cover either the structural or functional impairment. High variability in individual anatomy and physiology complicates studies on structure–function association [4,23,24,25]. Morphological studies with structural MRI revealed a marked variance across individuals in the extent of age-related brain change [26]. Research on brain functioning produced inconclusive findings on the onset and rate of episodic memory loss in the elderly. Different inherited and lifestyle factors account for these results [27,28]. A link between structural and functional impairment has not yet been explicitly explained.

Hypothetically, different brain parts have a common pace of age-related structural changes in the normal ageing, while a disproportion in the regional atrophy indicates accelerated brain ageing. Another hypothesis of the current study is the specificity of brain morphometry findings regarding normal growing and maturation in childhood and adolescence. In analogy to late life, morphometric changes in the pediatric population also remain an object of ongoing studies [29]. For this reason, we want to create a descriptive model of volumetric changes in the brain across the life-span.

The aim of our review is to find the characteristic features of non-pathological development and decline in distinct brain structures and to work out a precise descriptive model of brain morphometry among age groups. Specifically, we want to determine whether diverse brain compartments develop and decline proportionally throughout life and to calculate the annual rate of change.

## 2. Materials and Methods

We will perform a meta-analysis of available findings and modelled age-related changes in the brain. This approach is advantageous over a retrospective analysis or prospective observation, especially when researchers want to establish populational norms. A meta-analysis serves as a reliable source of data if performed in accordance with the standardized methodology [30,31]. Another reason for this study design is the intent to overcome limitations of original studies such as a small sample size, a narrow age range of cohorts, etc.

This protocol will be prepared in accordance with the Preferred Reporting Items for Systematic Review and Meta-Analyses Protocol (PRISMA-P) and registered in the PROSPERO International database of prospectively registered systematic reviews (protocol No: CRD42022354112). The PRISMA-P checklist is included in the supporting material (Appendix A).

### 2.1. Eligibility Criteria

To perform the search, we will pre-define and list the inclusion and exclusion criteria for the literature (see Table 1). Given that we focus on the structural change of the healthy population, the articles reporting findings on the following pathologies will be excluded: impaired development, mental disorders, organic brain pathologies, injuries to the head, and neurodegenerative diseases. We will not limit the age and sex of the study participants. The literature sources will be required to report findings of in vivo MRI examinations of the total brain or specific brain structures with outcome measurements such as absolute or proportional change in volume and/or size (width, length, thickness). We will exclude reviews, case reports, theses, dissertations, conference abstracts, editorial letters, and protocol papers. Animal studies and interventional research will also be eliminated from the search query.

### 2.2. Information Sources and Strategy

A comprehensive systematic search for literature will be conducted using four biomedical databases: PubMed, Embase, Scopus, and Web of Science. The pre-search was performed in September–October 2022 in PubMed and its Medical Subject Headings (MeSH). The review is set to start in November 2023. The strategy will be updated ahead of the manuscript submission. The names of 112 brain structures will be used as the keywords for the search strategy. The structures will be segmented from the brain MRI with the FreeSurfer software [32]. We will look for combinations of all the key terms in the “title”, “abstract”, and “MeSH”/”thesaurus” fields. MeSH terms will include “brain”, “Magnetic Resonance Imaging”, “organ size,” “atrophy”, “aging”, and “age factors”. Our search will be limited to the papers published in English from 1990 to 2023. We will conduct hand screening of reference lists in the retrieved papers. All records identified in the literature search will be uploaded to the systematic review software Covidence for automatic deduplication and blinded screening. Reproducible search strings for all databases will be appended to the review (Appendix A).

### 2.3. Study Selection

Once the papers are uploaded to the Covidence software, two independent reviewers will subsequently screen the title/abstract and full text against the predefined criteria. A third reviewer will resolve the discrepancies identified with the software. The reasons for exclusion of full text records will be stored. A PRISMA flow diagram will be used to report the screening process.

### 2.4. Data Extraction

Two independent reviewers will extract the data from the final list of papers into a summary of findings table. These records will include the basic characteristics (the authors, country, publication year, journal, study design, and mean age of study participants) in addition to the targeted data (change in size and volume of a brain structure within a period of time). The third reviewer will resolve eventual discrepancies in the data collection.

### 2.5. Quality Assessment of Individual Studies

Two independent reviewers will critically appraise each individual study included into the analysis. They will assess the quality of individual studies with the Joanna Brigs Institute checklist for analytical cross-sectional studies [33]. The tool has 8 questions with multiple choice answers “yes”, “no”, “unclear”, and “not applicable”. If a study scores less than 4 “yes” answers, it will be excluded from the statistical analysis.

We will construct funnel plots for each brain region and visually assess them. In the diagrams, the effect size is plotted against the standard error of the effect size. Asymmetry of the graph indicates publication bias. In our review, the effect size corresponds to annual atrophy of a studied brain structure.

### 2.6. Data Analysis and Synthesis

Prior to statistical analysis, we will explore the heterogeneity level of the studies with the Higgins–Thompson $I^{2}$ test [34]. Potential sources of heterogeneity include the strength of the magnetic field of MRI scanners, the type of segmentation, sample size, and study design. If $I^{2}$ exceeds 75%, we will perform a narrative systematic review instead of the meta-analysis. The final manuscript will present the number of qualifying articles and give a description of the overall trend of structural changes in the brain. The systematic review will analyse the sample size, age, and sex of the participants, and it will derive average statistical data on annual changes in brain regions. The team statistician will calculate the following parameters: (1) the average annual pace of enlargement or atrophy of each brain structure and (2) the side-specific change of brain structures in size or volume. For the analysis, we will use descriptive statistics and machine learning.

We will calculate the mean value of the left and right side volumes to receive the average size. To compensate for the variability in head size, the data will be estimated on a normalised volume in percentage to the total intracranial volume. Afterwards, we will compute the percentage of relative change per year, which is the first derivative of the model divided by the initial value and provided in % per year. To model the lifelong evolution of volumetric data for specific brain areas, we will consider linear, quadratic, cubic, or higher degree equations (see Equations (Equation 1)–(Equation 4)). Scatterplots in Figure 1 depict the models that we built based on the results of a preliminary search for references covering age-related change in the hippocampal [2,35,36,37,38,39,40,41] and lateral ventricle volumes [2,3,42,43,44].
(1)$$Vol=\beta_{0}+\beta_{1}Age+\epsilon$$
(2)$$Vol=\beta_{0}+\beta_{1}Age+\beta_{2}Age^{2}+\epsilon$$
(3)$$Vol=\beta_{0}+\beta_{1}Age+\beta_{2}Age^{2}+\beta_{3}Age^{3}+\epsilon$$
(4)$$Vol=\beta_{0}+\beta_{1}Age+\beta_{2}Age^{2}+\ldots +\beta_{k}Age^{k},\;k=1,10.$$

We will also use hybrid models with exponential cumulative distributions for growth with the linear, quadratic, cubic, or higher degree equations (see Equations (Equation 5)–(Equation 7)).
(5)$$Vol=\beta_{4}\left(1-e^{-Age/\beta_{5}}\right)+\beta_{0}+\beta_{1}Age+\epsilon$$
(6)$$Vol=\beta_{4}\left(1-e^{-Age/\beta_{5}}\right)+\beta_{0}+\beta_{1}Age+\beta_{2}Age^{2}+\epsilon$$
(7)$$Vol=\beta_{4}\left(1-e^{-Age/\beta_{5}}\right)+\beta_{0}+\beta_{1}Age+\beta_{2}Age^{2}+\beta_{3}Age^{3}+\epsilon .$$

Then, we will select the model explaining most of the data with a minimum number of parameters. To identify the best one among the candidate models, we will use a Bayesian information criterion (see a scatterplot in Figure 2). Finally, we will assess the portion of brain atrophy in a specific brain area by calculating the percentage of atrophy, i.e., its absolute relative difference with the control model. As the control model, we will use the relative rate of change in the CSF volume, which is a marker of the total brain shrinkage. The study pipeline is shown in Figure 3.

## 3. Discussion

### 3.1. Establishing Descriptive Model for Brain Ageing

A vast amount of literature is available on the volumetric decline of specific brain regions and neurocognitive slowing. Still, a thorough descriptive model of normal brain ageing is missing. This can be attributed to the following limitations that are typical for recent studies. First, researchers investigated specific age groups, and they did not study changes throughout life. Second, different methodological approaches used by the authors may account for incompatible findings [45]. For example, some studies used the cross-sectional design, and others—used the longitudinal one. Third, individual variations in brain structure limit our ability to establish population norms. Fourth, genetics, environmental, lifestyle, and cultural distinctions contribute to pronounced difference in brain morphology among nations. The current systemic review and meta-analysis investigates brain development and ageing in the global population.

MRI-based neuroimaging studies with voxel- and surface-based brain morphometry can detect a tiny change of brain structure. With these techniques, bioengineers help clinicians to quantify cortical and subcortical grey matter atrophy in terms of volume loss, macro-morphological changes, and cortical thinning [46,47]. One can evaluate structural damage to the white matter with voxel-based morphometry [46]. Recent studies reported the following outcomes of brain atrophy in normal ageing: volumetric reduction in the cortex and the sudden shrinkage of neuronal networks [48]. The latter describes the way in which brain atrophy impairs structural and functional connectivity. However, the aforementioned studies failed to provide a precise descriptive model of structural changes in cognitively preserved individuals, and future research should address this shortcoming. The evolution of brain structure in different life periods is briefly discussed in the next paragraphs.

#### 3.1.1. Period of Development

The total brain volume trajectory is strongly associated with age and cognitive status both in children and the elderly [49,50]. Radiological findings may promote the early diagnostics of impaired neurodevelopment. Still, diagnostic criteria for autism and other neurodevelopmental disorders are not uniform, and the resources for proper examination are limited. Various intellectual, behavioral, and psychiatric disorders in children may result from abnormal cortical development. To identify atypical change during the maturation period, physicians need the normative values for the cortical brain structures in 1–6 year old healthy children. This is the peak brain development period. The early signs of behavioral and developmental disorders become apparent at this age [42]. Researchers try to find out which particular mechanisms of brain development result in dysfunction in socio-emotional and communication networks in infants and toddlers with autism [51,52].

#### 3.1.2. Period of Maturation

The dynamics of the structural brain change in middle-aged adults, and older adults resemble the trajectory of functional performance throughout life [45,53]. From a neurophysiological point of view, an increase in performance lasts up to the age of 40, and it is followed by a plateau in neurocognitive functioning in the midlife and a decline in older adults [54,55,56,57]. The speed and extent of change seems to accelerate with age. It remains unknown if the pace of age-related transformation is common for distinct brain regions or if atrophy is slightly more prominent in certain locations.

#### 3.1.3. Period of Decline

Brain atrophy accounts for morphometric and functional changes in physiological and accelerated ageing. At the microscopic level, the atrophy presents with glial, myelin, axonal and/or neuronal loss. Macroscopically, brain atrophy results in brain shrinkage and compensatory enlargement of cerebrospinal fluid spaces: the ventricles and the subarachnoid space. A ventricular volume trajectory shows a strong association with age, because it is a summary marker of atrophy of the grey and white matter [58,59]. Specific structural markers serve as radiologic signs of disease-related atrophy, e.g., the hippocampal volume trajectory is clearly associated with amyloid angiopathy [58,60]. However, the sensitivity of the quantitative radiomical markers is too low to use them in screening for neurodegeneration, and their specificity is insufficient for differentiation among dementia subtypes. Large-scale standardised studies do not provide a comprehensive outline of the normal volumetric decline. In the absence of such studies, physicians make subjective judgements on the pace of brain ageing. This increases the chance of late or false diagnosis. Meanwhile, the therapeutic efficiency of anti-amyloid drugs is significantly higher at early stages of Alzheimer’s disease, which also justifies the importance of research on normal and pathological brain ageing [61,62,63,64,65,66].

### 3.2. Developing Reference Norms with Meta-Analysis

The current article is a protocol of the future study. Once we complete the analysis, we will compare the findings with the results of studies on relevant issues. For the discussion, we will consider available systematic reviews covering the following research questions: (1) an atrophy rate of brain parts vulnerable to changes in neurodevelopmental delay and neurodegenerative disorders and (2) brain structural correlates of the aforementioned pathologies.

Relationships between environmental factors and brain structure are not the major research topic for the study. Still, the data on environmental risks may show the multidimensionality of the research question. The latter is not limited to brain changes across the life, but it also includes the impact of adverse life events, dietary patterns, physical exercise, and vitamin or mineral supplementation on cognitive function in children and the elderly [67,68,69,70,71]. The future manuscript will contain the results of the prospective meta-analysis discussed in the context of healthy and pathological transformations in the brain.

The necessity of the current systematic review arises from the absence of reliable markers of delayed neurodevelopment and accelerated ageing: no threshold criteria for abnormal annual change in the brain are established. Visual assessment of the MRI by the radiologist can confirm dementia diagnosis when cognitive decline becomes prominent. However, subtle changes in the brain remain misreported [21]. This evidences the necessity of supporting clinical decision making with the analysis of radiomical data or developing computer-aided decision tools for automatic image analysis.

Statistics for meta-analysis help to diminish the risk of unreliable research outcomes resulting from inter-study heterogeneity. The meta-analytic approach allows for combining evidence from numerous studies, thus enlarging the study sample, as well as consolidating complex and sometimes conflicting findings [72]. After receiving data from different populations, we will apply a random-effects model to minimise errors in data presentation [72]. The model assists in controlling for unobserved heterogeneity, which is constant over time and not correlated with independent variables.

Our meta-analysis will illustrate life-long trends in brain volumetric changes. We will use regression models to provide trendlines for the variables of interest [73,74]. The trendlines will reflect normative values for volumetric changes in brain parts. This approach will allow us to identify people at risk of brain disorders at a certain age [74]. A large individual deviation from the normative curves may indicate pathologic ageing.

We will focus on the studies of cognitively normal people and model the annual rate of change in their brains. A reason for narrowing the research question to the cognitively intact population is limitations of the concept of accelerated ageing, which should be critically reappraised. For example, it remains unknown whether neurodegeneration is a kind of accelerated ageing [75,76] or an outcome of pathological changes in the body [75,77,78,79].

When studying accelerated ageing, one should consider individual multi-component reserves in the brain. The structural reserve refers to the number of neurons and synapses, whereas the cognitive reserve determines the ability of the brain to cope with structural brain damage [80]. Supposedly, individual cognitive and structural reserves affect the quality of life in the elderly population and modulate the risk of developing Alzheimer’s disease and other types of dementia. Variance in cognitive abilities and amount of neurons and synapses hardens the determination of the precise age of the organism. Calculation of the biological brain age is tricky, because indicators of normal ageing are still missing and the methodology of assessing reserves is not standardised [81]. Hence, we should rather focus on non-pathologic brain ageing.

The objective of the current study totally fits the idea of ‘Precision Medicine’, which is a concept of personalizing disease prevention and treatment. The concept integrates advanced statistical analysis into routine assessment of clinical findings along with the environmental, social, and behavioral factors impacting the individual [82]. Currently, radiological reports have limited value due to the semi-quantitative evaluation of structural changes by radiologists. Diagnostic errors arise from technical limitations of scanners and misinterpretation by physicians [83,84]. Therefore, bioengineering should create reliable tools for computerised diagnostics and automatic analysis of imaging findings.

We aim to improve the situation by applying quantitative assessment of radiological findings (radiomics) into practice instead of keeping it exceptionally for research. Radiomical findings will support early diagnostics and clinical decision making, thus meeting the demand of time. Radiomics mine high-dimensional data on organ structure and correlate this information with age and clinical endpoints [85]. Modelling structural changes in the healthy cohort will promote future studies on risk stratification of neurodevelopmental / neurodegenerative diseases with radiomics. These studies are currently performed and published, but their clinical applicability remains low [23,24,25,53,55,86,87,88].

## 4. Conclusions

This meta-analysis will help modelling the populational curves of brain development and ageing. For performing a comprehensive assessment of life-long structural changes, we will take into analysis multiple morphometry parameters, e.g., volume, thickness, length and width of brain regions. A systematic review and meta-analysis is the preferable study design for the formulated research question. Combining data from cross-sectional studies and longitudinal observations will allow us to acquire more statistical data on structural changes in the brain. The theoretical value of the future study is the implementation of highly sensitive screening and quantitative assessment of individual risks, which fully fits the idea of ‘Precision Medicine’. The practical application of the study is establishing the reference norms that could be used in screening individuals for developmental delay and cognitive decline.

## Figures and Tables

**Figure 1 biomedicines-11-01999-f001:**
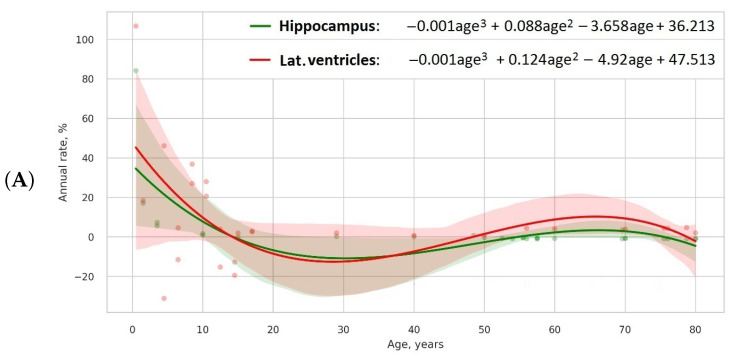
Cubic (**A**), fourth (**B**), and fifth order (**C**) models of lifelong change in hippocampal and lateral ventricle relative volumes.

**Figure 2 biomedicines-11-01999-f002:**
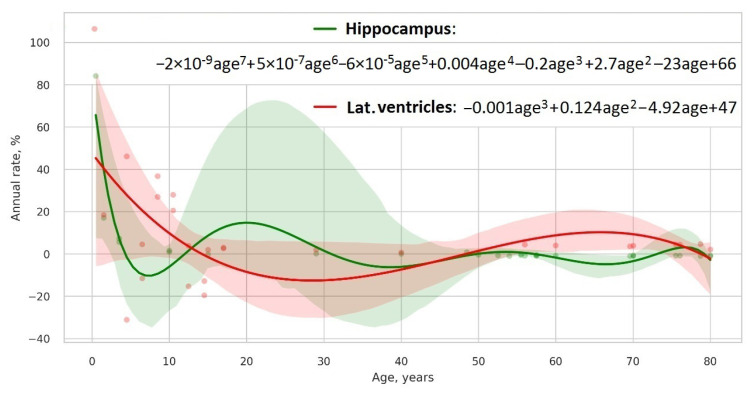
Best model types for hippocampus and lateral ventricles according to Bayesian information criterion.

**Figure 3 biomedicines-11-01999-f003:**
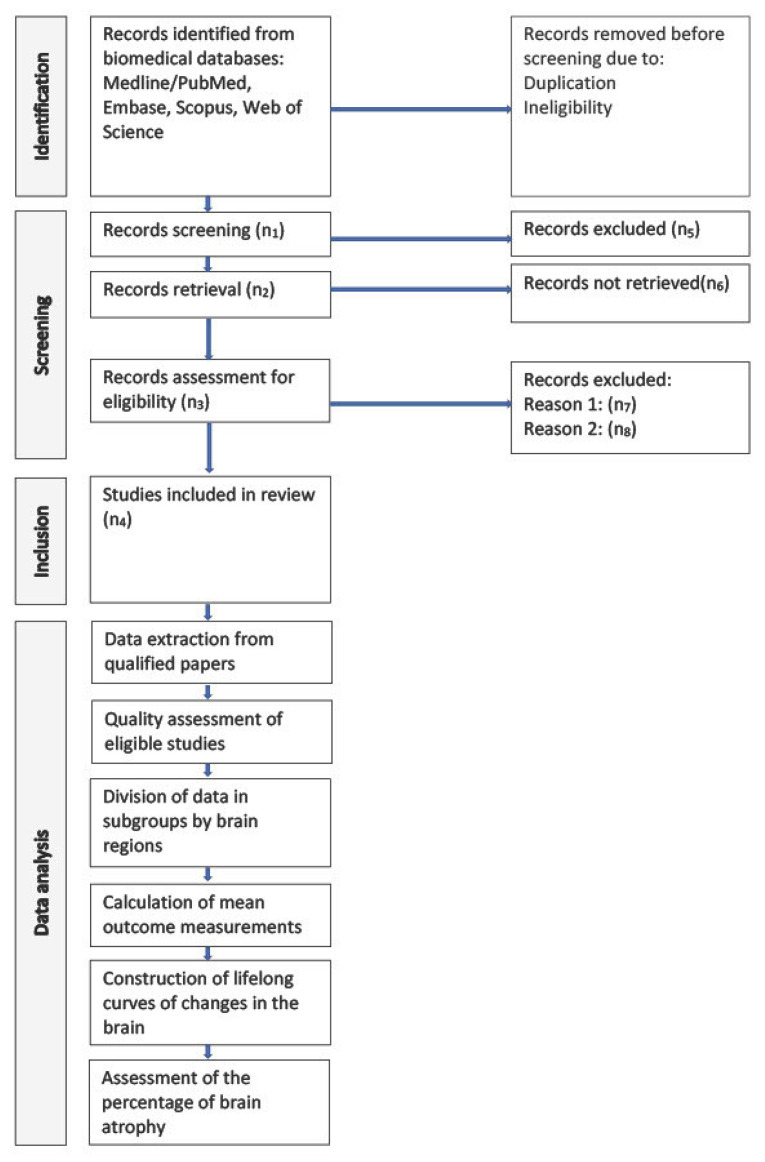
Study pipeline.

**Table 1 biomedicines-11-01999-t001:** Inclusion and exclusion criteria.

Inclusion Criteria	Exclusion Criteria	
	for Literature	for Subjects
1. Original peer-reviewed studies 2. Studies of the longitudinal and cross-sectional design 3. Studies on absolute or proportional change in volume, thickness, and other dimensions of the brain structures 4. Female and male participants of any age starting from birth 5. Individuals free from mental disorders, brain pathologies, and injuries	1. Grey literature 2. Editorial letters and protocol papers 3. Case studies and reviews 4. Studies performed on animals 5. Interventional studies (both therapeutic and surgical interventions) 6. Exposure of the participants to any factor that can potentially affect results.	Patients suffering from: 1. Mental and psychological disorders (F00–F99 in ICD-10) 2. Cerebrovascular diseases (I60–I69) 3. Organic pathology of the central nervous system (e.g., brain and meninges tumors: C71, D32–33) 4. Injury to the head (S00–S09)

## Data Availability

No datasets were generated or analysed during the current study. All relevant data from this study will be made available upon study completion.

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
