# Peer review of "Unraveling Lifelong Brain Morphometric Dynamics: A Protocol for Systematic Review and Meta-Analysis in Healthy Neurodevelopment and Ageing"

_biomedicines, 2023, doi:10.3390/biomedicines11071999_

Round 1

Reviewer 1 Report

We read the proposal for a systematic review titled “STRUCTURAL CHANGE IN THE BRAIN OVER A LIFESPAN: A PROTOCOL FOR SYSTEMATIC REVIEW AND META-ANALYSIS”. The authors are proposing searching for characteristic features of non-pathological development and degeneration in distinct brain structures and working out a precise descriptive model of brain morphometry in age groups. The work is to present a systematic review proposal where they are assessing MRI measurements of brains. They will calculate the volume or thickness of the brain structures assessing the prominence of atrophy in specific brain areas by adjusting the volumetric data to the total cerebrospinal fluid volume.

Comments:

The study design has major weaknesses as the output will be highly descriptive as many studies will have different parameters which the authors did not specify in their proposal.

What the authors are suggesting performing cannot be achieved via a systematic review as it requires the right controls adjusting for each age group.

The proposal does not include dates for the search of these research papers nor the MESH and keywords for searching for these keywords.

There are no inclusion and exclusion criteria.

what is described as assessing brain volume or thickness would require downloading the MRI images to compare, there is no way of normalizing the size.

There will be a lot of variation as several researchers would be using different kinds of MRIs and heterogeneity as the main variable.

Author Response

We read the proposal for a systematic review titled “STRUCTURAL CHANGE IN THE BRAIN OVER A LIFESPAN: A PROTOCOL FOR SYSTEMATIC REVIEW AND META-ANALYSIS”. The authors are proposing searching for characteristic features of non-pathological development and degeneration in distinct brain structures and working out a precise descriptive model of brain morphometry in age groups. The work is to present a systematic review proposal where they are assessing MRI measurements of brains. They will calculate the volume or thickness of the brain structures assessing the prominence of atrophy in specific brain areas by adjusting the volumetric data to the total cerebrospinal fluid volume.

Comments:

The study design has major weaknesses as the output will be highly descriptive as many studies will have different parameters which the authors did not specify in their proposal. What the authors are suggesting performing cannot be achieved via a systematic review as it requires the right controls adjusting for each age group.

We thank the reviewer for the comments on a rationale of the proposed methodology for investigating lifelong dynamics of brain development and degeneration. Considering a high volume of papers on the morphological change, we are certain that a meta-analysis will provide a pooled mean rate of changes in volume and thickness of brain structures. Other authors published systematic reviews and meta-analyses on similar topic in which they differentiated between morphological changes in the brains of healthy individuals and people with neurodegenerative or psychiatric (1,2). Our review focuses on individuals of any age free from mental, psychological, neurodegenerative disorders, organic pathologies of the central nervous system, and injuries to the head. We will not compare healthy individuals with the diseased population; therefore, we do not need to adjust for any controls. Instead of grouping studies by age groups, we will treat age as a continuous variable and adjust each studied cohort to the mean age of the study participants.

The proposal does not include dates for the search of these research papers nor the MESH and keywords for searching for these keywords.

Following the reviewer’s comments, we improved the “Information sources and strategy” section of the study. The following sentences were added:

Names of 112 brain structures will be used as the key words for the search strategy. The structures are segmented from the brain MRI with the FreeSurfer software. We will look for combinations of all the key terms in “title”, “abstract”, and “MeSH”/”thesaurus” fields. MeSH terms will include “brain”, “Magnetic Resonance Imaging”, “organ size,” “atrophy”, “aging”, and “age factors”. Our search will be limited to the papers published in English from 1990 to 2023.

There are no inclusion and exclusion criteria.

The inclusion and exclusion criteria are provided in Table 1 of the first version of the manuscript. While addressing reviewer’s comments, we revised the eligibility criteria for including individual studies into analysis. The new version of the section 2.1 states the following:

  To perform the search, we predefined and listed the inclusion and exclusion criteria for the literature (see Table 1). As we focus on the structural change of the healthy population, the articles reporting findings on the following pathologies will be excluded: impaired development, mental disorders, organic brain pathologies, injuries to the head, and neurodegenerative diseases. We will not limit age and sex of the study participants. The literature sources should report findings of in-vivo MRI examinations of the total brain or specific brain structures with the outcome measurements such as absolute or proportional change in volume and/or size (width, length, thickness). We will exclude reviews, case reports, theses, dissertations, conference abstracts, editorial letters, and protocol papers. Animal studies and interventional research will be also eliminated from the search query.

What is described as assessing brain volume or thickness would require downloading the MRI images to compare, there is no way of normalizing the size.

We understand the reviewer’s concern regarding possible methodological bias due to the brain size of participants in individual studies. The methodology of our meta-analysis is similar to the one described in the study by Fraser et al. on hippocampal atrophy in healthy people (3). In that review, authors extracted age of the participants, time between baseline and follow-up MRI examinations, segmentation type, annual atrophy rate for the total, right and left hippocampus. We will extract similar variables for all brain structures covered by the individual studies included into the analysis.

There will be a lot of variation as several researchers would be using different kinds of MRIs and heterogeneity as the main variable.

Inter-study heterogeneity is a common phenomenon in meta-analysis. We anticipate that strength of magnetic fields of MRI scanners, type of segmentation, sample size, and study design can be potential sources of heterogeneity. To identify a level of heterogeneity, we will calculate I2 index. If the value exceeds 75%, we will not do a meta-analysis. To minimize a potential methodological variability, we will perform a subgroup analysis where required.

Following reviewer’s comment, we updated sections (1) “Quality assessment of individual studies” and (2) “Data analysis and synthesis”.

(1) Two independent reviewers will critically appraise each individual study included into the analysis. They will assess the quality of individual studies with the Joanna Brigs Institute checklist for analytical cross-sectional studies. The tool has 8 questions with multiple choice answers “yes”, “no”, “unclear”, and “not applicable”. If a study scores less than 4 “yes” answers, it will be excluded from the statistical analysis.

We will construct funnel plots for each brain region and visually assess them. In the diagrams, the effect size is plotted against the standard error of the effect size. Asymmetry of the graph indicates publication bias. In our review, the effect size corresponds to annual atrophy of a studied brain structure.

(2) Prior to statistical analysis, we will explore heterogeneity level of the studies with the Higgins-Thompson I2 test. Potential sources of heterogeneity include the strength of magnetic field of MRI scanners, type of segmentation, sample size and study design. If I2 exceeds 75%, we will perform a narrative systematic review instead of the meta-analysis

Reviewer 2 Report

The authors present the results of a systematic review examining and evaluating those data which agree and suggest markers of age-associated changes in the brain. Many of the changes associated with neurological disorders revolve around age-associated changes, or accelerated mechanisms regarding such. Therefore, reaching a consensus on those markers may yield targets which should be prioritised for investigation towards diagnostic and therapeutic purposes. This reviews aims to examine and scrutinise the existing literature in an effort to compile those changes which represent the most viable targets to examine. Overall, this is a well-constructed piece that is clear in its objectives and outcomes for the most part.  

In reviewing the manuscript, I made a couple of observations. The following should be considered by the authors when preparing a suitable revision.

1.       In Figure 3, it would be useful if the numbers of articles evaluated at each step were included for reference.

2.       In Figures 1 and 2, the legends could be made clearer by increasing the font and moving the text to outside of the graph area itself.

Author Response

The authors present the results of a systematic review examining and evaluating those data which agree and suggest markers of age-associated changes in the brain. Many of the changes associated with neurological disorders revolve around age-associated changes, or accelerated mechanisms regarding such. Therefore, reaching a consensus on those markers may yield targets which should be prioritised for investigation towards diagnostic and therapeutic purposes. This reviews aims to examine and scrutinise the existing literature in an effort to compile those changes which represent the most viable targets to examine. Overall, this is a well-constructed piece that is clear in its objectives and outcomes for the most part.  

In reviewing the manuscript, I made a couple of observations. The following should be considered by the authors when preparing a suitable revision.

In Figure 3, it would be useful if the numbers of articles evaluated at each step were included for reference.

We thank the reviewer for a positive comment and would like to highlight that the current paper is a proposal of a methodology of the upcoming systematic review. Figure 3 illustrates the steps needed to address the objectives of the study. The selection process of individual studies will be depicted in PRISMA-flowchart which will be constructed automatically in the Covidence software. 

In Figures 1 and 2, the legends could be made clearer by increasing the font and moving the text to outside of the graph area itself.

We appreciate reviewer’s comments and suggestions to improve the quality of the figures. The new Figures are uploaded along with the edited manuscript.

Reviewer 3 Report

The current review is about brain structural changes occurring during aging. The annual rate of change was calculated for the volume or thickness of the brain structures. They tried to model the lifelong dynamics of brain measurements by using machine learning. They also assessed the prominence of atrophy in specific brain areas by adjusting the volumetric data to the total cerebrospinal fluid volume. It is a very interesting study but there are several concerns remained to be addressed:

1.     Authors mentioned that the aim of this review is to find the characteristic features of non-pathological development and degeneration in distinct brain structures and to work out a precise descriptive model of brain morphometry in age groups. In that case the title of review is not representative of the aim of study. Authors need to change the title to be more related to the aim and findings of the review.

2.     Since the current review is supposed to be helpful for understanding the brain structural changes over the age, it is important to be more comprehensive. One of the important factors is to classify the studies and findings based on the clinical and preclinical studies and perform modelling based on this classification rather than pooling them together. That would give great information to the neuroscientist regarding the animal and clinical studies.

3.     It is very important for the preclinical studies authors provide modeling based on the animal models (mice, rats,…) , since aging will be different between them.

4.     Authors only focused on the hippocampus and lateral ventricle, however, in the title, they mentioned brain in that case they need to include more brains regions which are important such as cerebellum, substantial negra, prefrontal cortex. Or they should change the title and mentioning specifically hippocampus and LV.

5.     One of the important confounding factors is a sex , so it is important that authors modelling the structural changes in the brain based on the sex as well.

Author Response

The current review is about brain structural changes occurring during aging. The annual rate of change was calculated for the volume or thickness of the brain structures. They tried to model the lifelong dynamics of brain measurements by using machine learning. They also assessed the prominence of atrophy in specific brain areas by adjusting the volumetric data to the total cerebrospinal fluid volume. It is a very interesting study but there are several concerns remained to be addressed:

Authors mentioned that the aim of this review is to find the characteristic features of non-pathological development and degeneration in distinct brain structures and to work out a precise descriptive model of brain morphometry in age groups. In that case the title of review is not representative of the aim of study. Authors need to change the title to be more related to the aim and findings of the review.

The authors appreciate the reviewer’s suggestion to change the article title. To make it descriptive of the study aim, we propose the following heading:

“Modelling lifelong dynamics of morphometric changes in brain parts in healthy neurodevelopment and ageing: protocol for systematic review and meta-analysis”

Since the current review is supposed to be helpful for understanding the brain structural changes over the age, it is important to be more comprehensive. One of the important factors is to classify the studies and findings based on the clinical and preclinical studies and perform modelling based on this classification rather than pooling them together. That would give great information to the neuroscientist regarding the animal and clinical studies.  It is very important for the preclinical studies authors provide modeling based on the animal models (mice, rats,…) , since aging will be different between them.

We thank for the suggestion to include preclinical studies in the review and run it in parallel with the analysis of the human data. To take this into consideration, we conducted a preliminary literature search with the search strings listed in Supplemental Material. The preliminary search returned a total number of 3775 relevant papers in a single database. This suggest narrowing the research question rather than extending it. However, we decided to proceed with the current aim because we will study 112 brain parts separately. The inclusion of animal beings in the search criteria will make the results abundant and hard to process and interpret.  

Authors only focused on the hippocampus and lateral ventricle, however, in the title, they mentioned brain in that case they need to include more brains regions which are important such as cerebellum, substantial negra, prefrontal cortex. Or they should change the title and mentioning specifically hippocampus and LV.

The current paper describes objectives and methodology which will be applied to the prospective meta-analysis. We modeled life-long dynamics of relative volumes of the hippocampus and lateral ventricles to illustrate possible outcomes of the final meta-analysis. We aim to construct trendlines for the whole brain and distinct brain structures.

One of the important confounding factors is a sex, so it is important that authors modelling the structural changes in the brain based on the sex as well.

We thank the reviewer for a suggestion to model structural changes in the brain for the total population, men, and women separately. This type of subgroup analysis will be possible if we receive a sufficient amount of papers.

Round 2

Reviewer 1 Report

accept

Author Response

We Thank the reviewer for his decision to accept the manuscript.

Reviewer 3 Report

Authors replied to the comments by explanation for each comment.

Author Response

The authors appreciate the reviewer's feedback.